# An Attribute-based Method for Video Anomaly Detection

**Tal Reiss**                                                                *tal.reiss@mail.huji.ac.il*
*School of Computer Science and Engineering*
*The Hebrew University of Jerusalem, Israel*

**Yedid Hoshen**                                                            *yedid.hoshen@mail.huji.ac.il*
*School of Computer Science and Engineering*
*The Hebrew University of Jerusalem, Israel*

**Reviewed on OpenReview:** *https://openreview.net/forum?id=XL1N6iLrOG*

## Abstract

Video anomaly detection (VAD) identifies suspicious events in videos, which is critical for crime prevention and homeland security. In this paper, we propose a simple but highly effective VAD method that relies on attribute-based representations. The base version of our method represents every object by its velocity and pose, and computes anomaly scores by density estimation. Surprisingly, this simple representation is sufficient to achieve state-of-the-art performance in ShanghaiTech, the most commonly used VAD dataset. Combining our attribute-based representations with an off-the-shelf, pre-trained deep representation yields state-of-the-art performance with a $99.1\%, 93.7\%$, and $85.9\%$ AUROC on Ped2, Avenue, and ShanghaiTech, respectively. Our code is available at https://github.com/talreiss/Accurate-Interpretable-VAD.

## 1 Introduction

Video anomaly detection (VAD) aims to discover interesting but rare events in video. The task has attracted much interest as it is critical for crime prevention and homeland security. One-class classification (OCC) is one of the most popular VAD settings, where the training set consists of normal videos only, while at test time the trained model needs to distinguish between normal and anomalous events. The key challenge for learning-based VAD is that classifying an event as normal or anomalous depends on the human operator's particular definition of normality. Differently from supervised learning, there are no training examples of anomalous events, this essentially requires the learning algorithms to have strong priors.

Many previous VAD methods use a combination of deep networks with self-supervised objectives. A popular line of work consists of training a neural network to predict the next frame and classifying it as anomalous if the predicted and observed frames differ significantly. While such methods can achieve good performance, they do not make their priors explicit i.e., it is unclear why they consider some frames more anomalous than others, making them difficult to debug and improve. More recent methods involve a preliminary object extraction stage from video frames, implying that objects are significant for VAD. However, they typically use object-level self-supervised approaches that do not make their priors explicit. Our hypothesis is that making priors more explicit will improve VAD performance.

In this paper, we propose a new approach that directly represents each video frame by simple attributes that are semantically meaningful to humans. Our method extracts objects from every frame, and represents each object by two attributes: its velocity and body pose (in the case the object is human). These attributes are well known to be important for VAD (Markovitz et al., 2020; Georgescu et al., 2021a). We detect anomalous values of these representations by using density estimation. Concretely, our method classifies a frame as anomalous if it contains one or more objects that have unusual values of velocity and/or pose (see Fig. 1). Our simple velocity and pose representations achieve state-of-the-art performance (85.9% AUROC) on the most popular VAD dataset, ShanghaiTech.

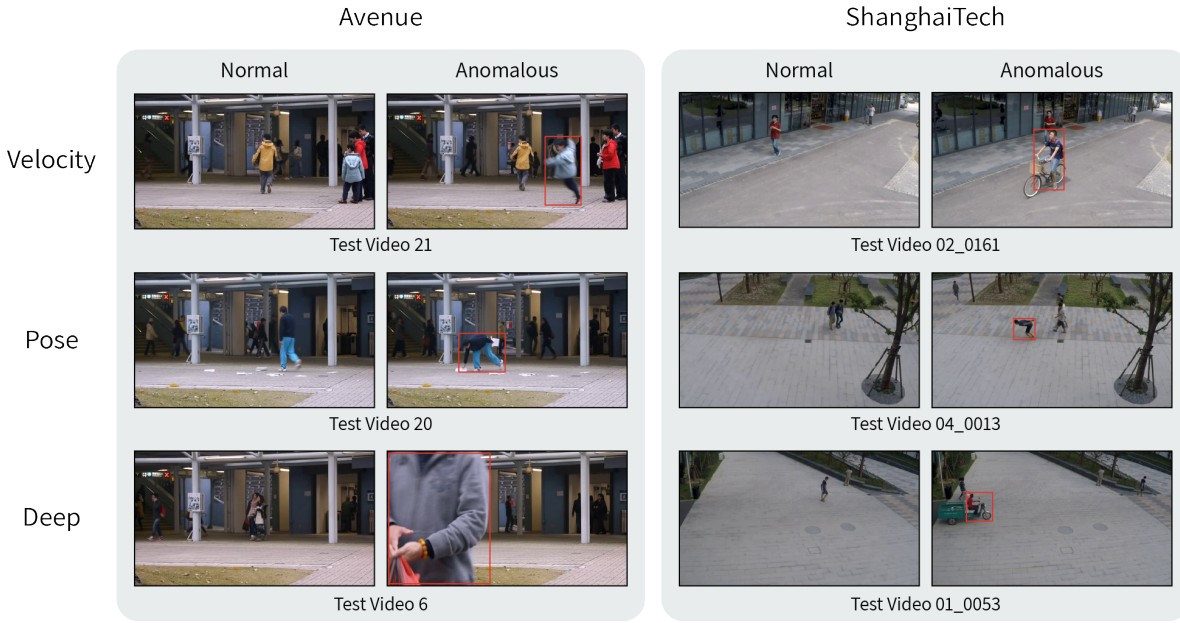

Figure 1: **The Avenue and ShanghaiTech datasets.** We present the most normal and anomalous frames for each feature. For anomalous frames, we visualize the bounding box of the object with the highest anomaly score. Best viewed in color.

While the velocity and pose representations are highly effective, they ignore other attributes, most importantly object category. As an example, if we never see a lion in normal videos, a test video containing a lion is anomalous. To represent these *residual* attributes, we model them with an off-the-shelf, deep representation (here, we use CLIP features). Our final method combines velocity, pose and the deep representations. It achieves state-of-the-art performance on the three most commonly reported datasets.

## 2 Related Work

Classical video anomaly detection methods were typically composed of two steps: handcrafted feature extraction and anomaly scoring. Some of the manual features that were extracted were: optical flow histograms (Chaudhry et al., 2009; Colque et al., 2016; Perš et al., 2010) and SIFT (Lowe, 2004). Commonly used scoring methods include: density estimation (Eskin et al., 2002; Glodek et al., 2013; Latecki et al., 2007), reconstruction (Jolliffe, 2011), and one-class classification (Scholkopf et al., 2000).

In recent years, deep learning has gained in popularity as an alternative to these early works. The majority of video anomaly detection methods utilize at least one of three paradigms: reconstruction-based, prediction-based, skeletal-based, or auxiliary classification-based methods.

**Reconstruction & prediction based methods.** In the reconstruction paradigm, the normal training data is typically characterized by an autoencoder, which is then used to reconstruct input video clips. The assumption is that a model trained solely on normal training clips will not be able to reconstruct anomalous frames. This assumption does not always hold true, as neural networks can often generalize to some extent out-of-distribution. Notable works are (Nguyen & Meunier, 2019; Chang et al., 2020; Hasan et al., 2016b; Luo et al., 2017b; Yu et al., 2020; Park et al., 2020).

Prediction-based methods learn to predict frames or flow maps in video clips, including inpainting intermediate frames, predicting future frames, and predicting human trajectories (Liu et al., 2018a; Feng et al., 2021b; Chen et al., 2020; Lee et al., 2019; Lu et al., 2019; Park et al., 2020; Wang et al., 2021; Feng et al., 2021a; Yu et al., 2020). Additionally, some works take a hybrid approach combining the two paradigms (Liu et al., 2021b; Zhao et al., 2017; Ye et al., 2019; Tang et al., 2020; Morais et al., 2019). As these methods are

trained to optimize both objectives, input frames with large reconstruction or prediction errors are considered anomalous.

**Self-supervised auxiliary tasks.** There has been a great deal of research on learning from unlabeled data. A common approach is to train neural networks on suitably designed auxiliary tasks with automatically generated labels. Tasks include: video frame prediction (Mathieu et al., 2016), image colorization (Zhang et al., 2016; Larsson et al., 2016), puzzle solving (Noroozi & Favaro, 2016), rotation prediction (Gidaris et al., 2018), arrow of time (Wei et al., 2018), predicting playback velocity (Doersch et al., 2015), and verifying frame order (Misra et al., 2016). Many video anomaly detection methods use self-supervised learning. In fact, self-supervised learning is a key component in the majority of reconstruction-based and prediction-based methods. SSMTL (Georgescu et al., 2021a) trains a CNN jointly on three auxiliary tasks: arrow of time, motion irregularity, and middle-box prediction, in addition to knowledge distillation. Jigsaw-Puzzle (Wang et al., 2022) trains neural networks to solve spatio-temporal jigsaw puzzles. The networks are then used for VAD.

**Skeletal methods.** Such methods rely on a pose tracker to extract the skeleton trajectories of each person in the video. Anomalies are then detected using the skeleton trajectory data. Our attribute-based method outperforms previous skeletal methods (e.g., (Markovitz et al., 2020; Rodrigues et al., 2020; Yu et al., 2021; Sun & Gong, 2023)) by a large margin. In concurrent work, (Hirschorn & Avidan, 2023) proposed a skeletal-based method that achieved comparable results to ours on the ShanghaiTech dataset but was outperformed by large margins on the rest of the evaluated benchmarks. Different from skeletal approaches, our method does not require pose tracking, which is extremely challenging in crowded scenes. Our pose features only use a single frame, while our velocity features only require a pair of frames. In contrast, skeletal approaches require pose tracking across many frames, which is expensive and error-prone. It is also important to note that skeletal features by themselves are ineffective in detecting non-human anomalies, therefore, being insufficient for providing a complete VAD solution.

**Object-level video anomaly detection.** Early methods, both classical and deep learning, operated on entire video frames. This proved difficult for VAD as frames contain many variations, as well as a large number of objects. More recent methods (Georgescu et al., 2021a; Liu et al., 2021b; Wang et al., 2022) operate at the object level by first extracting object bounding boxes using off-the-shelf object detectors. Then, they detect if each object is anomalous. This is an easier task, as objects contain much less variation than whole frames. Object-based methods yield significantly better results than frame-level methods.

It is often believed that due to the complexity of realistic scenes and the variety of behaviors, it is difficult to craft features that will discriminate between them. As object detection was inaccurate prior to deep learning, classical methods were previously applied at the frame level rather than at the object level, and therefore underperformed on standard benchmarks. We break this misconception and demonstrates that it is possible to craft semantic features that are accurate.

## 3 Method

### 3.1 Preliminaries

Our method assumes a training set consisting of $N_c$ video clips $\{c_1, c_2 ... c_{N_c}\} \in \mathcal{X}_{train}$ that are all normal (i.e., do not contain any anomalies). Each clip $c_i$ is comprised of $N_i$ frames, $c_i = [f_{i,1}, f_{i,2}, ... f_{i,N_i}]$. The goal is to classify each frame $f \in c$ in an inference clip $c$ as normal or anomalous. Our method represents each frame $f$ as $\phi(f) \in \mathbb{R}^d$, where $d \in \mathbb{N}$ is the feature dimension. We compute the anomaly score of frame $f$ using an anomaly scoring function $s(\phi(f))$, and classify it as anomalous if $s(\phi(f))$ exceeds a threshold.

### 3.2 Overview

We propose a method that represents each video frame as a set of objects, with each object characterized by its attributes. This contrasts with most previous methods that do not explicitly represent attributes. Specifically, we focus on two key attributes: velocity and body pose. Object velocity is probably the most important attribute as it can detect if an object is moving unusually fast e.g., running away, or in a strange

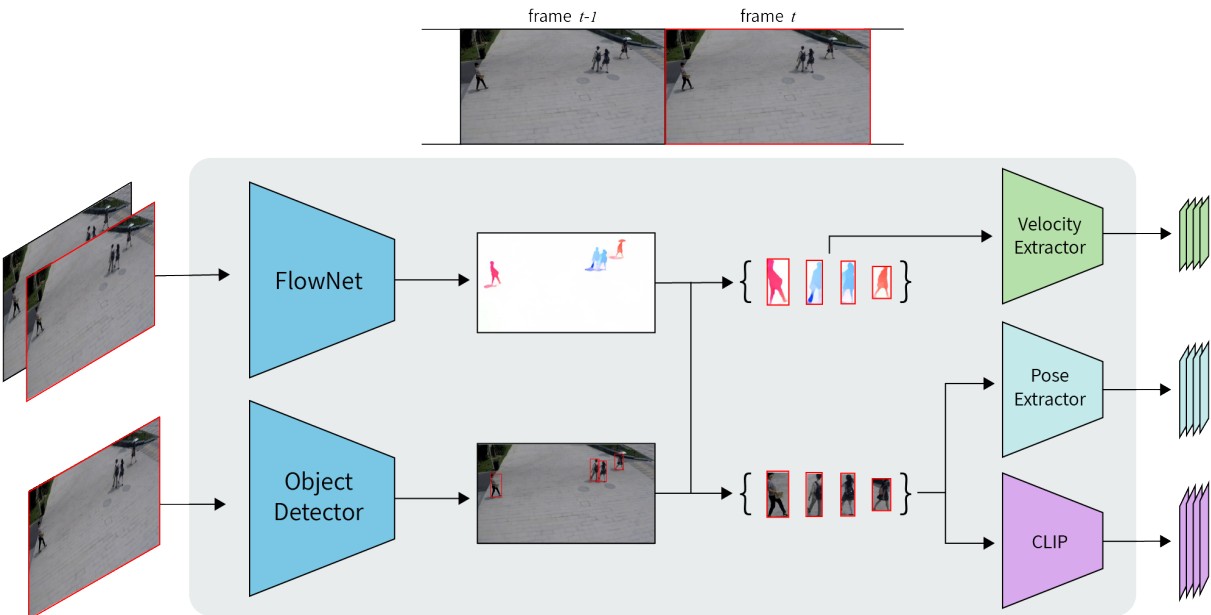

Figure 2: **An overview of our method.** We first extract optical flow maps and bounding boxes for all of the objects in the frame. We then crop each object from the original image and its corresponding flow map. Our representation consists of velocity, pose, and deep (CLIP) features.

direction e.g., against the direction of traffic. Given the challenges of capturing true 3D velocity without depth information, we use optical flow as an effective proxy to represent apparent motion between frames. Similarly, human body poses can reveal anomalous activities, such as throwing an item. Instead of using the full 3D human pose, we represent it with 2D landmarks, which are easier to detect. Both attributes have been used in previous VAD studies, such as Georgescu et al. (2021a) and Markovitz et al. (2020), as part of more complex approaches.

We compute the anomaly score based on density estimation of object-level feature descriptors. This is done in three stages: pre-processing, feature extraction, and density estimation. In the pre-processing stage, our method (i) uses an off-the-shelf motion estimator to estimate the optical flow for each frame; (ii) localizes and classifies the bounding boxes of all objects within a frame using an off-the-shelf object detector. The outputs of these models are used to extract object-level velocity, pose, and deep representations (see Sec. 3.4). Finally, our method uses density estimation to calculate the anomaly score of each test frame. See Fig. 2 for an illustration.

### 3.3 Pre-processing

Our method extracts velocity and pose from each object in each video frame. To do so, we compute optical flow and body landmarks for each object in the video.

**Optical flow.** Our method uses optical flow as a proxy for its velocity. We extract the optical flow map $o$ for each frame $f \in c$ in every video clip $c$ using an off-the-shelf optical flow model.

**Object detection.** Our method models frames by representing every object individually. This follows many recent papers, e.g., (Georgescu et al., 2021a; Liu et al., 2021b; Wang et al., 2022) that found that object-based representations are more effective than global, frame-level representations. We first detect all objects in each frame using an off-the-shelf object detector. Formally, our object detection generates a set of $m$ bounding boxes $bb_1, bb_2...bb_m$ for each frame, with corresponding category labels $y_1, y_2, ..., y_m$.

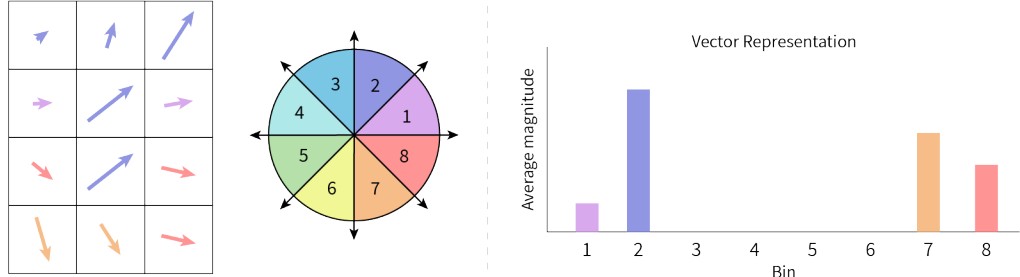

Figure 3: **An illustration of our velocity feature vector.** *Left:* We quantize the orientations into $B = 8$ equi-spaced bins, and assign each optical flow vector in the object's bounding box is to a single bin. *Right:* The value of each bin is the average magnitude of the optical flow vectors assigned to this bin. Best viewed in color.

### 3.4 Feature extraction

Our method represents each object by two semantic attributes, velocity and pose, and by an implicit, deep representation.

**Velocity features.** Our working hypothesis is that an unusual speed or direction of motion can often identify anomalies in video. As objects can move in both $x$ and $y$ axes and both the magnitude (speed) and orientation of the velocity may be anomalous, we compute velocity features for each object in each frame. We begin by cropping the frame-level optical flow map $o$ by the bounding box $bb$ of each detected object. Following this step, we obtain a set of cropped object flow maps (see Fig. 2). We rescale these flow maps to a fixed size of $H_{flow} \times W_{flow}$. Next, we represent each flow map with the average motion for each orientation, where orientations are quantized into $B \in \mathbb{N}$ equi-spaced bins, following classical optical flow representations e.g., (Chaudhry et al., 2009). The final representation is a $B$-dimensional vector consisting of the average flow magnitude of the flow vectors in each bin (see Fig. 3). This representation is capable of describing motion in both radial and tangential directions. We denote our velocity feature extractor as: $\phi_{velocity} : H_{flow} \times W_{flow} \to \mathbb{R}^B$.

**Pose features.** Most anomalous events in video involve humans, so we include activity features in our representation. While a full understanding of activity requires temporal features, we find that human body pose from a single frame can detect many unusual activities. Even though human pose is essentially a 3D feature, 2D body landmark positions already provide much of the signal. We compute pose features for each human object $bb$ using an off-the-shelf 2D keypoint extractor that outputs the pixel coordinates of each landmark position, denoted by $\hat{\phi}_{pose}(bb) \in \mathbb{R}^{2 \times d}$, where $d \in \mathbb{N}$ is the number of keypoints. To ensure invariance to the human's position and size, we normalize the keypoints. First, we subtract the coordinates of the top-left corner of the object bounding box from each landmark. We then scale the $x$ and $y$ axes so that the object bounding box has a fixed size of $H_{pose} \times W_{pose}$, where $H_{pose}$ and $W_{pose}$ are constants. Formally, let $l \in \mathbb{R}^2$ be the top-left corner of the human bounding box. The pose descriptor becomes:

$$\phi_{pose}(bb) = \begin{pmatrix} \frac{H_{pose}}{height(bb)} & 0 \\ 0 & \frac{W_{pose}}{width(bb)} \end{pmatrix} (\hat{\phi}_{pose}(bb) - l) \tag{1}$$

Here, $height(bb)$ and $width(bb)$ are the height and width of the object bounding box $bb$, respectively and $l \in \mathbb{R}^2$ be the top-left corner of $bb$. Finally, we flatten $\phi_{pose}$ to obtain the final pose feature vector.

**Deep features.** While velocity is the most discriminative attribute, other attributes beyond pose may also matter. Deep features implicitly bundle together many different attributes. Hence, we use them to model the residual attributes which are not described by velocity and pose. We follow previous anomaly detection works (Reiss et al., 2021; Reiss & Hoshen, 2023), that used generic, pretrained encoders to implicitly represent image attributes. Concretely, we use a pretrained CLIP encoder (Radford et al., 2021), $\phi_{deep}(.)$, to represent the bounding box of each object in each frame. Note that CLIP representations do not achieve competitiveness

on their own; in fact, they perform much worse than the velocity representations (see Tab. 3). However, together, velocity, pose, and CLIP features represent video sufficiently well to outperform the state-of-the-art.

### 3.5 Density Estimation

We use density estimation for scoring samples as normal or anomalous, where a low estimated density indicates anomaly. To estimate density, we fit a separate estimator for each feature. As velocity features are low-dimensional, we use a Gaussian mixture (GMM) estimator. As our pose and deep features are high-dimensional, we estimate their density using $k$NN. Specifically, we compute the $L_2$ distance between the feature $x$ of a target object and the nearest $k$ exemplars in the corresponding training feature set. We compare different exemplar selection methods is in Sec. 4.5. We denote our density estimators by $s_{vel}(.), s_{pose}(.), s_{deep}(.)$.

**Score calibration.** Combining the three density estimators requires calibration. To do so, we estimate the distribution of anomaly scores on the normal training set. We then scale the scores using min-max normalization. The $k$NN used for scoring pose and deep features present a subtle point. When computing $k$NN on the training set, the exemplars must not be taken from the same clip as the target object. The reason is that the same object appears in nearby frames with virtually no variation, distorting $k$NN estimates. Instead, we compute the $k$NN between each training set object and all objects in the other video clips provided in the training set. We can now define $\forall f \in \{velocity,\ pose,\ deep\}$: $\mu_f = \max_x\{s_f(\phi_f(x))\}$ and $\nu_f = \min_x\{s_f(\phi_f(x))\}$.

### 3.6 Inference

We extract a representation for each object $bb$ in each frame $f$ in each inference clip, as describe above. We then compute an anomaly score for each attribute feature of each object $bb$. The score for every frame is simply the maximum score across all objects. The final anomaly score is the sum of the individual feature scores normalized by our calibration parameters:

$$s(f) = \max_k\Big\{\frac{s_{vel}(\phi_{velocity}(bb_k)) - \nu_{velocity}}{\mu_{velocity} - \nu_{velocity}}\Big\} + \max_k\Big\{\frac{s_{pose}(\phi_{pose}(bb_k)) - \nu_{pose}}{\mu_{pose} - \nu_{pose}}\Big\} +$$
$$\max_k\Big\{\frac{s_{deep}(\phi_{deep}(bb_k)) - \nu_{deep}}{\mu_{deep} - \nu_{deep}}\Big\} \quad (2)$$

As anomalous events span multiple frames, we smooth the frame scores using a temporal smoothing filter.

## 4 Experiments

### 4.1 Datasets

We evaluated our method on three publicly available VAD datasets, using their training and test splits. Only test videos included anomalous events. We report the statistics of the datasets in Tab. 1.

**UCSD Ped2.** This dataset (Mahadevan et al., 2010) contains 16 normal training videos and 12 test videos at a $240 \times 360$ pixel resolution. Videos show a fixed scene with a camera above the scene and pointed downward. The training video clips contain only normal behavior of pedestrians walking, while examples of abnormal events are bikers, skateboarding, and cars.

**CUHK Avenue.** This dataset (Lu et al., 2013) contains 16 normal training videos and 21 test videos at $360 \times 640$ pixel resolution. Videos show a fixed scene using a ground-level camera. Training video clips contain only normal behavior. Examples of abnormal events are strange activities (e.g. throwing objects, loitering, and running), movement in the wrong direction, and abnormal objects.

**ShanghaiTech Campus.** This dataset (Liu et al., 2018a) is the largest publicly available dataset for VAD. There are 330 training videos and 107 test videos from 13 different scenes at $480 \times 856$ pixel resolution. ShanghaiTech contains video clips with complex light conditions and camera angles, making this dataset

Table 1: Statistics of the evaluation datasets.

| Dataset | Number of Frames | | | | | Scenes | Anomaly Types |
|---|---|---|---|---|---|---|---|
| | Total | Train set | Test set | Normal | Anomalous | | |
| UCSD Ped2 | 4,560 | 2,550 | 2,010 | 2,924 | 1,636 | 1 | 5 |
| CUHK Avenue | 30,652 | 15,328 | 15,324 | 26,832 | 3,820 | 1 | 5 |
| ShanghaiTech | 317,398 | 274,515 | 42,883 | 300,308 | 17,090 | 13 | 11 |

more challenging than the other two. Anomalies include robberies, jumping, fights, car invasions, and bike riding in pedestrian areas.

## 4.2 Implementation Details

We use ResNet50 Mask-RCNN (He et al., 2017) pretrained on MS-COCO (Lin et al., 2014) to extract object bounding boxes. To filter out low confidence objects, we follow the same configurations as in (Georgescu et al., 2021a). Specifically for Ped2, Avenue, and ShanghaiTech, we set confidence thresholds of 0.5, 0.8, and 0.8. In order to generate optical flow maps, we use FlowNet2 (Ilg et al., 2017). For our landmark detection, we use AlphaPose (Fang et al., 2017) pretrained on MS-COCO with $d = 17$ keypoints. We use a pretrained ViT B-16 CLIP (Dosovitskiy et al., 2020; Radford et al., 2021) image encoder as our deep feature extractor. Our method is built around the extracted objects and flow maps. We use $H_{velocity} \times W_{velocity} = 224 \times 224$ to rescale flow maps. As for $H_{pose} \times W_{pose}$ rescaling, we calculate the average height and width from the bounding boxes of the train set and use those values. The lower resolution of Ped2 prevents objects from filling a histogram, and to extract pose representations, therefore we use $B = 1$ orientations and rely solely on velocity and deep representations. We use $B = 8$ orientations for Avenue and ShanghaiTech. When testing, for anomaly scoring we use $k$NN for the pose and deep representations with $k = 1$ nearest neighbors. For velocity, we use GMM with $n = 5$ Gaussians. Finally, the anomaly score of a frame represents the maximum score among all the objects within that frame.

## 4.3 Evaluation Metrics

Our study uses standard VAD evaluation metrics. We vary the threshold over the anomaly scores to measure the frame-level Area Under the Receiver Operation Characteristic (AUROC) with respect to the ground-truth annotations. We report two types of AUROC: (i) micro-averaged AUROC, which computes the score by on all frames from all videos; (ii) macro-averaged AUROC, which computes the AUROC score individually for each video and then averages the scores of all videos. Most existing studies report micro-averaged AUROC, while only a few report macro-averaged AUROC.

## 4.4 Quantitative Results

We compare our method and the state-of-the-art from recent years in Tab. 2. We took the performance numbers of the baseline methods directly from the original papers.

**Ped2 results.** Most methods obtained over 94% on Ped2, indicating that of the three public datasets, it is the simplest. While our method is comparable to the current state-of-the-art method (HF$^2$ (Liu et al., 2021b)) in terms of performance, the near-perfect results on Ped2 indicate it is practically solved.

**Avenue results.** Our method obtained a new state-of-the-art micro-averaged AUROC of 93.7%. Our method also outperformed the current state-of-the-art in terms of macro-averaged AUROC by a considerable margin of 2.8%, reaching 96.3%.

**ShanghaiTech results.** Our method outperforms all previous methods on the largest dataset, ShanghaiTech, by a considerable margin. Accordingly, our method achieves 85.9% AUROC, higher than the best performance previous methods achieved, 85.1% (MS-VAD (Zhang et al., 2024)). We note that in concurrent

Table 2: **Frame-level AUROC (%) comparison.** The best and second-best results are bolded and underlined, respectively.

| Year | Method | Ped2 | | Avenue | | ShanghaiTech | |
|---|---|---|---|---|---|---|---|
| | | Micro | Macro | Micro | Macro | Micro | Macro |
| ≤ 2019 | HOOF (Chaudhry et al., 2009) | 61.1 | - | - | - | - | - |
| | HOFM (Colque et al., 2016) | 89.9 | - | - | - | - | - |
| | SCL (Lu et al., 2013) | - | - | 80.9 | - | - | - |
| | Conv-AE (Hasan et al., 2016a) | 90.0 | - | 70.2 | - | - | - |
| | StackRNN (Luo et al., 2017a) | 92.2 | - | 81.7 | - | 68.0 | - |
| | STAN (Lee et al., 2018) | 96.5 | - | 87.2 | - | - | - |
| | MC2ST (Liu et al., 2018b) | 87.5 | - | 84.4 | - | - | - |
| | Frame-Pred. (Liu et al., 2018a) | 95.4 | - | 85.1 | - | 72.8 | - |
| | Mem-AE. (Gong et al., 2019) | 94.1 | - | 83.3 | - | 71.2 | - |
| | CAE-SVM (Ionescu et al., 2019) | 94.3 | 97.8 | 87.4 | 90.4 | 78.7 | 84.9 |
| | BMAN (Lee et al., 2019) | 96.6 | - | 90.0 | - | 76.2 | - |
| | AM-Corr (Nguyen & Meunier, 2019) | 96.2 | - | 86.9 | - | - | - |
| 2020 | MNAD-Recon. (Park et al., 2020) | 97.0 | - | 88.5 | - | 70.5 | - |
| | CAC (Wang et al., 2020) | - | - | 87.0 | - | 79.3 | - |
| | Scene-Aware (Sun et al., 2020) | - | - | 89.6 | - | 74.7 | - |
| | VEC (Yu et al., 2020) | 97.3 | - | 90.2 | - | 74.8 | - |
| | ClusterAE (Chang et al., 2020) | 96.5 | - | 86.0 | - | 73.3 | - |
| 2021 | AMMCN (Cai et al., 2021) | 96.6 | - | 86.6 | - | 73.7 | - |
| | SSMTL (Georgescu et al., 2021a) | 97.5 | 99.8 | 91.5 | 91.9 | 82.4 | 89.3 |
| | MPN (Lv et al., 2021) | 96.9 | - | 89.5 | - | 73.8 | - |
| | HF$^2$ (Liu et al., 2021a) | **99.3** | - | 91.1 | 93.5 | 76.2 | - |
| | CT-D2GAN (Feng et al., 2021a) | 97.2 | - | 85.9 | - | 77.7 | - |
| | BA-AED (Georgescu et al., 2021b) | 98.7 | 99.7 | 92.3 | 90.4 | 82.7 | 89.3 |
| 2022 | SSPCAB (Ristea et al., 2022) | - | - | 92.9 | 91.9 | 83.6 | 89.5 |
| | DLAN-AC (Yang et al., 2022) | 97.6 | - | 89.9 | - | 74.7 | - |
| | Jigsaw-Puzzle (Wang et al., 2022) | 99.0 | **99.9** | 92.2 | 93.0 | 84.3 | **89.8** |
| 2023 | USTN-DSC (Yang et al., 2023) | 98.1 | - | 89.9 | - | 73.8 | - |
| | EVAL (Singh et al., 2023) | - | - | 86.0 | - | 76.6 | - |
| | FB-SAE (Cao et al., 2023) | 97.1 | 99.2 | 86.8 | 89.1 | 79.2 | 80.2 |
| | FPDM (Yan et al., 2023) | - | - | 90.1 | - | 78.6 | - |
| | LMPT (Shi et al., 2023) | 97.6 | - | 90.9 | - | 78.8 | - |
| | STF-NF (Hirschorn & Avidan, 2023) | 93.1 | 91.2 | 60.1 | 63.5 | **85.9** | 87.8 |
| 2024 | SD-MAE (Ristea et al., 2024) | 95.4 | 98.4 | 91.3 | 90.9 | 79.1 | 84.7 |
| | MS-VAD (Zhang et al., 2024) | - | - | 92.4 | 92.9 | 85.1 | **89.8** |
| | Ours | 99.1 | **99.9** | **93.7** | **96.3** | **85.9** | 89.6 |

work, STF-NF (Hirschorn & Avidan, 2023) achieved comparable results to ours on the ShanghaiTech dataset. Our method outperforms it by large margins on Ped2 (by 6.0% AUROC) and Avenue (by 33.6% AUROC).

To summarize, our method achieves the highest performance on the three most popular public benchmarks. It simply consists of three simple representations and does not require training.

## 4.5 Analysis

**Ablation study.** We report in Tab. 3 the anomaly detection performance on the Ped2, Avenue and ShanghaiTech datasets of all attribute combinations. Our findings reveal that the velocity features provide the highest frame-level AUROC on Ped2, Avenue and ShanghaiTech, with 98.8%, 86.0% and 84.4% micro-averaged AUROC, respectively. In ShanghaiTech, our velocity features *on their own* are already state-of-the-art com-

Table 3: **Ablation study.** Result are in frame-level AUROC (%). The best and second-best results are in bold and underline, respectively.

| Pose Features | Deep Features | Velocity Features | Ped2 | | Avenue | | ShanghaiTech | |
|:---:|:---:|:---:|:---:|:---:|:---:|:---:|:---:|:---:|
| | | | Micro | Macro | Micro | Macro | Micro | Macro |
| ✓ | | | - | - | 73.8 | 76.2 | 74.5 | 81.0 |
| | ✓ | | 96.4 | 95.3 | 85.4 | 87.7 | 72.5 | 82.5 |
| | | ✓ | 98.8 | 99.6 | 86.0 | 89.6 | 84.4 | 84.8 |
| ✓ | ✓ | | - | - | 89.3 | 88.8 | 76.7 | 84.9 |
| | ✓ | ✓ | **99.1** | **99.9** | 93.0 | 95.5 | 84.5 | 88.7 |
| ✓ | | ✓ | - | - | 86.8 | 93.0 | **85.9** | 88.8 |
| ✓ | ✓ | ✓ | - | - | **93.7** | **96.3** | 85.1 | **89.6** |

Table 4: Comparison of different numbers of velocity features bins ($B$). Frame-level AUROC (%) results. Best in bold.

| Bins ($B$) | Avenue | | ShanghaiTech | |
|:---|:---:|:---:|:---:|:---:|
| | Micro | Macro | Micro | Macro |
| $B = 1$ | 83.5 | 83.5 | 81.2 | 80.9 |
| $B = 2$ | 84.1 | 83.8 | 82.1 | 82.7 |
| $B = 4$ | 85.5 | 89.2 | 84.0 | 84.6 |
| $B = 8$ | **86.0** | **89.6** | **84.4** | **84.8** |
| $B = 16$ | 84.1 | 88.4 | 83.1 | 84.2 |

Table 5: Our final results when $k$NN is replaced by $k$-means. Frame-level AUROC (%). Time is expressed in average ms per frame. Best in bold.

| $k =$ | Avenue | | | ShanghaiTech | | |
|:---|:---:|:---:|:---:|:---:|:---:|:---:|
| | Mic. | Mac. | Time | Mic. | Mac. | Time |
| 1 | 91.8 | 94.0 | 0.51 | 84.2 | 87.2 | 0.45 |
| 5 | 92.0 | 94.2 | 0.52 | 84.3 | 88.1 | 0.45 |
| 10 | 92.1 | 94.5 | 0.52 | 84.6 | 88.1 | 0.45 |
| 100 | 92.9 | 95.2 | 0.53 | 84.8 | 88.6 | 0.46 |
| All | **93.7** | **96.3** | 4.93 | **85.1** | **89.6** | 36.0 |

pared with all previous VAD methods. We expect this to be due to the large number of anomalies associated with speed and motion, such as running people and fast-moving objects, e.g. cars and bikes. Adding either pose or CLIP improved performance, mostly macro-AUROC, presumably as it provided information about human activity which accounts for some of the anomalies in this dataset. Velocity features were still the most performant on Ped2 and Avenue. However, combining them with deep features improved performance significantly. Overall, we observe that using all three features performed the best on Avenue. Due to the extremely low resolution of the Ped2 dataset, pose feature extraction is not feasible, so we rely solely on velocity and deep features for this dataset.

**Number of velocity bins.** We ablated the impact of different numbers of bins ($B$) in our velocity features in Tab. 4. We compared AUROC scores on the Avenue and ShanghaiTech datasets. The results indicate that the choice of $B$ influences detection accuracy. Specifically, we observed that increasing the number of bins from $B = 1$ to $B = 8$ led to consistent improvements in both micro and macro AUROC scores on both datasets. This suggests that a finer quantization of velocity orientations represents motion better and improves anomaly detection. Performance gains diminish beyond $B = 8$.

**$k$-Means as a faster alternative.** Computing $k$NN has linear complexity in the number of objects in the datasets, which may be slow for large datasets. We can speed it up by reducing the number of samples via $k$-means. In Tab. 5, we compare the performance of our method with $k$NN and $k$-means. Note that $k$-means still uses $k$NN to calculate anomaly scores as the sum of distances to nearest neighbor means. This is much faster than the original $k$NN as there are fewer means than the number of objects in the training set. We observe that it improves inference time with a small accuracy loss.

**Pose features for non-human objects.** We extract pose representations exclusively for human objects and not for non-human objects. We calculate the pose anomaly score for each frame by taking the score of the object with the most anomalous pose. Non-human objects are given a pose anomaly score of $-\infty$ and therefore do not contribute to the frame-wise pose anomaly score. While we acknowledge that non-human objects can also exhibit anomalies, our method leverages velocity and deep representations to capture these types of events.

Table 6: Comparison of FlowNet2 vs. RAFT for flow map extraction. Frame-level AUROC (%) based on velocity features.

| Backbone | Avenue | | ShanghaiTech | |
|---|---|---|---|---|
| | Micro | Macro | Micro | Macro |
| RAFT | 85.7 | 89.7 | 84.3 | 84.2 |
| FlowNet2 | 86.0 | 89.6 | 84.4 | 84.8 |

Table 7: Comparison of Mask R-CNN vs. YOLO-v8 for object detection. Frame-level AUROC (%) based on velocity features.

| Backbone | Avenue | | ShanghaiTech | |
|---|---|---|---|---|
| | Micro | Macro | Micro | Macro |
| YOLO-v8 | 84.8 | 87.4 | 83.1 | 82.7 |
| Mask-RCNN | 86.0 | 89.6 | 84.4 | 84.8 |

Table 8: Comparison of video encoders and CLIP. Frame-level AUROC (%) results. Best in bold.

| Encoder | Level | Avenue | | ShanghaiTech | |
|---|---|---|---|---|---|
| | | Micro | Macro | Micro | Macro |
| TimeSformer (Bertasius et al., 2021) | Frame | 61.2 | 64.1 | 58.2 | 60.1 |
| TimeSformer (Bertasius et al., 2021) | Object | 63.1 | 64.0 | 59.1 | 59.2 |
| VideoMAE V2 (Wang et al., 2023) | Frame | 68.3 | 67.9 | 60.3 | 60.5 |
| VideoMAE V2 (Wang et al., 2023) | Object | 67.0 | 68.1 | 60.0 | 59.9 |
| DINO | Object | 77.6 | 81.2 | 71.2 | 80.3 |
| CLIP (ours) | Object | **85.4** | **87.7** | **72.5** | **82.5** |

**Backbone analysis.** We performed additional ablation studies to evaluate the impact of different backbone networks on the overall performance of our method. Specifically, we tested alternative backbones for optical flow (FlowNet2 vs. RAFT (Teed & Deng, 2020)), as shown in Tab. 6 and object detection (Mask R-CNN vs. YOLO-v8) in Tab. 7. The results indicate that the effectiveness of our approach is primarily driven by the feature design rather than any specific choice of backbone.

**Why do we use an image encoder instead of a video encoder?** Recent self-supervised learning methods such as TimeSformer (Bertasius et al., 2021), VideoMAE (Tong et al., 2022; Wang et al., 2023), X-CLIP (Ni et al., 2022), and CoCa (Yu et al., 2022) have significantly improved the performance of pretrained video encoders on downstream tasks like Kinetics-400 (Kay et al., 2017). It is therefore reasonable to expect that video encoders, which capture both temporal and spatial information, would outperform image encoders in video anomaly detection (VAD). However, in our experiments, we found that features extracted by pretrained video encoders did not perform as well as those extracted from pretrained image encoders on VAD benchmark datasets. We hypothesize that this weaker performance is due to the video encoders' focus on capturing frame-level temporal dynamics, whereas our method is object-centric. Additionally, when we tested video encoders on 10-frame windows of fixed object bounding boxes (centered around time $t$), we observed no performance gain, likely due to resolution constraints and the need for high-quality contextual information. Tab. 8 summarizes our findings on the limited effectiveness of video encoders in this setting. Additionally, we evaluated DINO (Caron et al., 2021) as a comparison to CLIP and found that while DINO performed slightly worse than CLIP, it still outperformed video encoders. This result, with DINO showing only a slight performance drop compared to CLIP, demonstrates that our deep features are not dependent on a specific image encoder.

**Running times.** We carried out all our experiments on a NVIDIA RTX 2080 GPU. Our preprocessing stage, which includes object detection and optical flow extraction, takes approximately 80 milliseconds (ms) per frame. It takes our method approximately 5 ms to compute the velocity extraction, pose extraction, and deep features extraction stages, combined with anomaly scoring. Our method runs at 12FPS with an average of 5 objects per frame. For comparison, we evaluated two other methods on the same hardware: BA-AED (Georgescu et al., 2021b) runs at 24 FPS, while HF[2] Liu et al. (2021a), 2021) runs at 12 FPS. Our method's running speed is comparable to HF[2] but slightly slower than BA-AED.

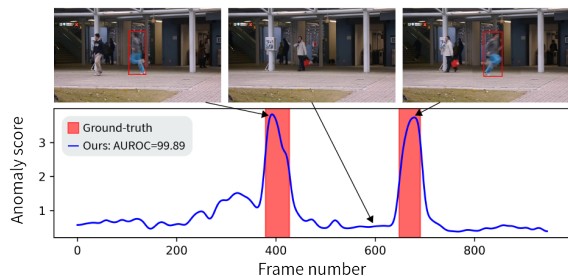

Figure 4: Frame-level scores and anomaly localizations for Avenue's test video 04. Best viewed in color.

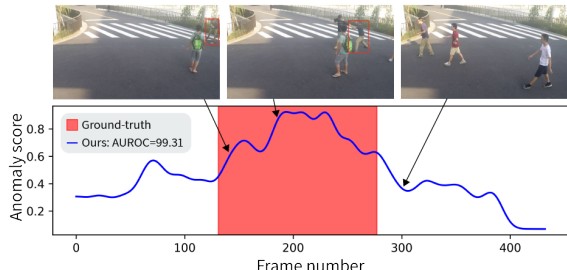

Figure 5: Frame-level scores and anomaly localizations for ShanghaiTech's test video 03_0059. Best viewed in color.

### 4.6 Qualitative Results

We visualize the anomaly detection process for Avenue and ShanghaiTech in Fig. 4 and Fig. 5, where we plot the anomaly scores across all frames of a video. Our anomaly scores are clearly highly correlated with anomalous events, demonstrating the effectiveness of our method.

## 5 Discussion

**Exploring other semantic attributes.** There are other important attributes for VAD beyond velocity and pose. Identifying other relevant attributes that correlate with anomalous events can further improve anomaly detection systems. For example, attributes related to object interactions, spatial arrangements, or temporal patterns may be very discriminative for some types of anomalies. Finding ways to systematically discover such attributes may significantly speed up research.

**Guidance.** Finding relevant attributes for anomaly detection may require user guidance. In real-world scenarios, operators have domain knowledge about factors that lead to anomalous behavior. Efficiently incorporating this guidance, such as selecting velocity or pose features in our work, is essential for leveraging this knowledge effectively.

**Other academic benchmarks.** While our method, using simple attributes, was effective on the three most popular VAD datasets, extending it to more complex datasets may require more work. Publicly available datasets such as UCF-Crime (Sultani et al., 2018) and XD-Violence (Wu et al., 2020), which feature a wider variety of anomalies and larger scales, present additional challenges. These datasets are essentially different from the ones tested here as they contain *distinct* scenes in training and testing data, and include moving cameras, which also change the scene. So far, only weakly-supervised VAD has been successful on these datasets as they labeled anomalous data in training. The field needs new, more complex datasets within the fixed camera setting to further stress-test one-class classification VAD methods such as ours.

## 6 Ethical Considerations

While VAD offers significant potential for enhancing public safety and security, it is crucial to acknowledge and address the ethical implications of such technology. VAD systems, including our proposed method, can be used in surveillance applications, which raises important privacy concerns. The continuous monitoring of public spaces may lead to a sense of constant observation, potentially infringing on individuals' right to privacy and freedom of movement. Moreover, there is a risk that VAD systems could be misused for unauthorized tracking or profiling of individuals.

To mitigate these ethical risks, several strategies should be considered in VAD systems development and deployment. First, strict data protection protocols should be implemented to ensure that collected video data is securely stored, accessed only by authorized individuals, and deleted after a defined period. Second, VAD use should be transparent, with clear warnings informing individuals when they are entering areas

under surveillance. Third, VAD systems should be designed with privacy-preserving techniques, such as immediate data anonymization or the use of low-resolution data that can detect anomalies without identifying individuals. By implementing these measures, we can work towards harnessing VAD technology benefits while respecting individual privacy and civil rights.

In addition to technical safeguards, it is also necessary to consider regulatory and oversight mechanisms to ensure responsible deployment. We recommend that VAD systems be subject to civilian oversight, where independent authorities evaluate their use, especially in sensitive contexts like law enforcement or public monitoring. Such oversight would help prevent potential misuse, ensuring that VAD systems are applied in ways that benefit society without compromising human rights. Furthermore, restrictions should be placed on VAD deployment for purposes other than public safety, with guidelines that limit its use to specific cases where the benefits clearly outweigh the risks. These guidelines could include requiring legal authorization for certain VAD uses, particularly in private spaces or in applications that extend beyond standard anomaly detection use-cases.

## 7 Conclusion

We propose a simple yet highly effective attribute-based method for video anomaly detection (VAD). Our method represents each object in each frame using velocity and pose representations and uses density estimation to compute anomaly scores. These simple representations are sufficient to achieve state-of-the-art performance on the ShanghaiTech and Ped2 datasets. By combining attribute-based representations with implicit deep representations, we achieve top VAD performance with AUROC scores of 99.1%, 93.7%, and 85.9% on Ped2, Avenue, and ShanghaiTech, respectively. Our extensive ablation study highlights the relative merits of the three representations. Overall, our method is both accurate and easy to implement.

## Acknowledgment

This research was partially supported by funding from the Israeli Science Foundation and the KLA Corporation. Tal Reiss is supported by the Google Fellowship and the Israeli Council for Higher Education.

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

# A    More Qualitative Results

We provide more qualitative results of our methods over the evaluation datasets.

In Ped2, Fig. 6 and Fig. 7 demonstrate the effectiveness of our method, which can easily detect fast-moving objects such as trucks and bicycles. Accordingly, we can conclude that Ped2 has been practically solved based on the near-perfect results obtained by our method (as well as many others). Fig. 8 shows that our method is capable of detecting anomalies within a short timeframe. Fig. 9 and Fig. 10 provide more qualitative information regarding our method's ability to detect anomalies of various types. In this way, our method achieves a new state-of-the-art in Avenue and ShanghaiTech, surpassing other approaches by a wide margin.

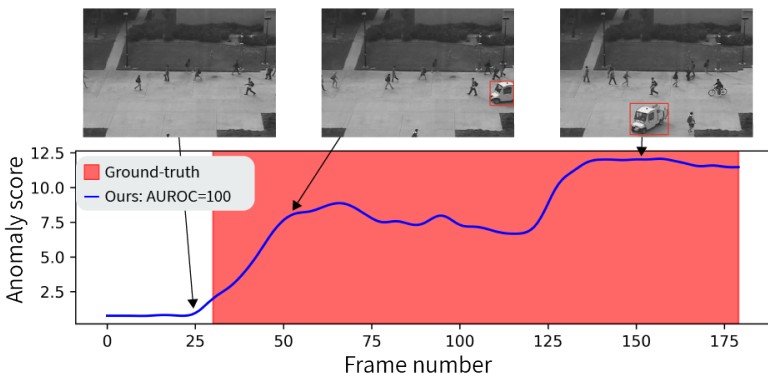

Figure 6: Frame-level scores and anomaly localization examples for test video 04 from Ped2.

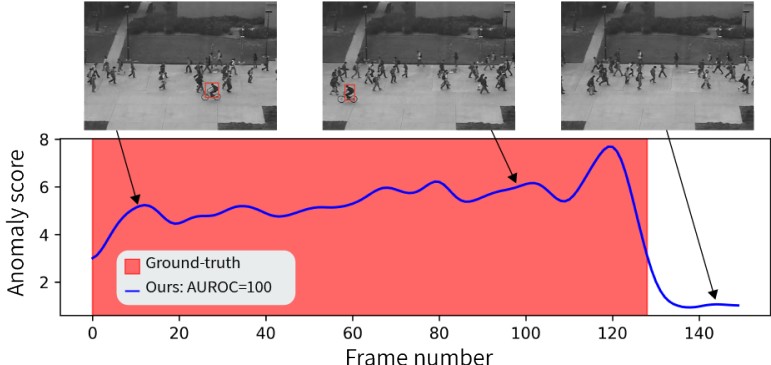

Figure 7: Frame-level scores and anomaly localization examples for test video 05 from Ped2.

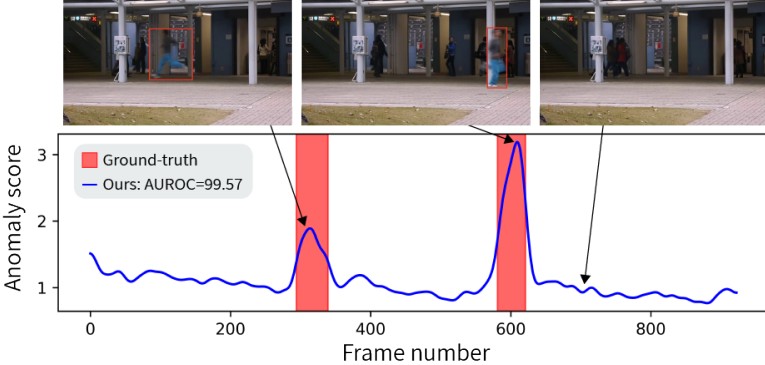

Figure 8: Frame-level scores and anomaly localization examples for test video 03 from Avenue.

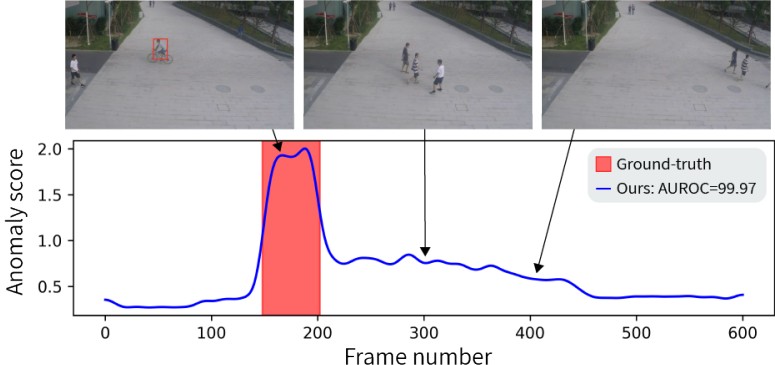

Figure 9: Frame-level scores and anomaly localization examples for test video 01_0025 from ShanghaiTech.

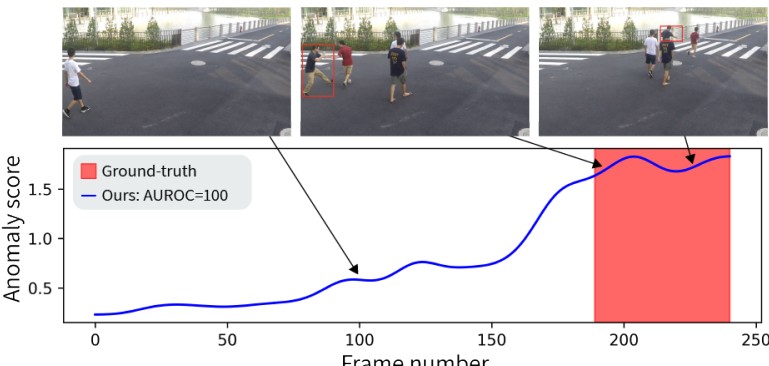

Figure 10: Frame-level scores and anomaly localization examples for test video 07_0048 from ShanghaiTech.

## B   More Analysis & Discussion

**Per-scene breakdown.** In Tab. 9, we present the per-scene performance of our method on the ShanghaiTech dataset, which is the only dataset among the three benchmarks that includes multiple scenes. The results demonstrate that our method performs consistently well across most scenes, with a few exceptions. Specifically, scenes 8, 9, and 10 demonstrate lower performance compared to others. These scenes (i.e., videos with the prefix 08_**, 09_**, 10_**) feature anomalies involving complex activities such as erratic jumping, throwing objects, and pushing people, as well as frequent occlusions. Such activities involve high

Table 9: ShanghaiTech per-scene frame-level micro AUROC (%) results.

| Scene Number | Total Test Frames | Total Test Anomalies | Anomaly Ratio | AUROC |
|---|---|---|---|---|
| 01 | 11,894 | 4,884 | 0.41 | 88.2 |
| 02 | 1,155 | 662 | 0.57 | 87.8 |
| 03 | 4,090 | 1,212 | 0.29 | 90.8 |
| 04 | 4,761 | 1,874 | 0.39 | 87.0 |
| 05 | 4,160 | 1,016 | 0.24 | 94.6 |
| 06 | 1,470 | 702 | 0.47 | 94.5 |
| 07 | 3,368 | 886 | 0.26 | 92.7 |
| 08 | 3,708 | 1,992 | 0.53 | 67.8 |
| 09 | 361 | 84 | 0.23 | 72.6 |
| 10 | 2,213 | 1,539 | 0.69 | 64.1 |
| 11 | 337 | 141 | 0.41 | 99.3 |
| 12 | 3,74 | 2,334 | 0.71 | 81.1 |

levels of motion and interaction between multiple subjects, which likely challenges the velocity-based feature representations, leading to reduced performance.

**What are the benefits of pretrained features?** Previous anomaly detection works (Reiss et al., 2021; Reiss & Hoshen, 2023; Reiss et al., 2022; 2024) demonstrated that using feature extractors pretrained on external, generic datasets (e.g. ResNet on ImageNet classification) achieves high anomaly detection performance. This was demonstrated on a large variety of datasets across sizes, domains, resolutions, and symmetries. These representations achieved state-of-the-art performance on distant domains, such as aerial, microscopy, and industrial images. As the anomalies in these datasets typically had nothing to do with velocity or human pose, it is clear the pretrained features model many attributes beyond velocity and pose. Consequently, by combining our attribute-based representations with CLIP's image encoder, we are able to emphasize both explicit attributes (velocity and pose) derived from real-world priors and attributes that cannot be described by them, allowing us to achieve the best of both worlds.

