# OpenReview forum: "An Attribute-based Method for Video Anomaly Detection"
_TMLR — Accepted by TMLR_

### Review · Reviewer_zhpF · 2024-08-25

**Summary Of Contributions:**

In this submission, the authors proposed a simple but effective video anomaly detection method based on the attributes of visual objects in video. In particular, the authors found that in existing datasets, most anomaly scenarios are associated with anomaly pose, velocity, and depth of human beings. Accordingly, the authors combined a flow-based model with a visual detector, extracting representations for the three attributes. Based on these representations, the proposed video anomaly detector achieves higher accuracy in existing datasets.

**Audience:**

Yes

**Claims And Evidence:**

Yes

**Requested Changes:**

See the weaknesses above.

**Strengths And Weaknesses:**

Strengths:

1. The paper is well-written and easy to follow.

2. The experimental part is solid, including sufficient baselines, ablation studies, and hyperparameter analysis.


Weaknesses:

1. My main concern is the generalizability of the proposed method. In my opinion, the anomaly scenarios in a video can be associated with non-human objects. In addition, for both human and non-human objects, attributing their anomaly behaviors purely to their poses, velocities, and depths may be over-simplified. As a result, although the proposed method achieves encouraging performance in the existing datasets, the tasks associated with these datasets are too simple to reflect real-world video anomaly detection problems. Actually, many baselines can also achieve high performance (>90% accuracy), which verifies my concern to some extent.

---

> ### Author Response · Authors · 2024-10-06
>
> Thank you for your review. We appreciate the reviewer’s recognition of our paper's clarity and the thoroughness of our experimental work. Regarding the reviewer’s concern about generalizability, we acknowledge that real-world video anomaly detection scenarios can indeed involve non-human objects and more complex anomalies. Our method does consider non-human objects, primarily through velocity and deep features, while pose features are specific to humans. We agree that attributing anomalies solely to pose and velocity is a simplification. However, our approach demonstrates that even these basic attributes can capture a significant portion of anomalous events in current benchmarks. The deep features in our method serve as a residual for other potential attributes, allowing some flexibility in detecting diverse anomalies.
>
> The reviewer raises an important point about the simplicity of existing datasets. We agree that these benchmarks may not fully reflect the complexity of real-world VAD problems. In fact, this limitation of current datasets is precisely why we included a discussion on the need for more challenging benchmarks in our paper. Our method's strong performance on these datasets, while encouraging, is indeed partly a reflection of their limitations. However, we believe our approach of using explicit attributes provides a valuable foundation for tackling more complex scenarios. As the field progresses towards more challenging datasets, our method's modularity allows for easy incorporation of additional attributes or more sophisticated representations as needed. We see our work as a step towards more interpretable and adaptable VAD systems, rather than a final solution. Future work could explore incorporating a wider range of attributes and testing on more diverse, real-world scenarios to further validate and extend our approach.

---

### Review · Reviewer_z24s · 2024-09-30

**Summary Of Contributions:**

This paper proposes a method for Video Anomaly Detection (VAD; a binary classification task to detect whether each frame in a given video is an anomaly or not) based on a frame representation that combines several different features focusing on object attributes such as velocity and pose, along with deep representations from a pre-trained CLIP. Given this representation, a K-nearest neighbor method (for the pose and deep features) and a Gaussian Mixture Model (GMM; for the velocity feature) are used to compute the anomaly score as given in Equation 2, and the final decision is made if the score exceeds a threshold.

The experiment is designed to reveal the binary classification performance in terms of AUROC, and the authors claim that the proposed method leads to state-of-the-art performance on three major VAD benchmarks: Ped2, Avenue, and ShanghaiTech. As for the ablation study, different combinations of the three features (i.e., the velocity feature, the pose feature, and the CLIP feature) are tested, the influence of the number of bins hyperparameter in the velocity feature is studied, and finally, an alternative model that replaces the kNN method with a lightweight k-means is evaluated, which trade-offs some classification performance for a practical runtime speed of the method.

**Audience:**

Yes

**Broader Impact Concerns:**

I have no concerns about broader impacts.

**Claims And Evidence:**

Yes

**Requested Changes:**

Please refer to the weaknesses, and provide further explanations or possibly the update in the manuscript.

**Strengths And Weaknesses:**

**(Strength) The method is simple yet effective.**
The proposed method leverages features readily available from the off-the-shelf models, such as optical flows (velocity) and object detectors (pose), and CLIP (a deep feature). Although the method has a simple design to combine the anomaly scores from each feature, it still achieves state-of-the-art performance in the experiment.

**(Weakness) The motion attribute design solely depending on optical flows.**
The proposed method relies on velocity features based on optical flows, which might be effective for (single) object-centric anomalies, but might struggle in more complex scenarios, such as those involving interactions between multiple objects causing occlusions. In such a case, the optical flow method may introduce significant errors in the motion attribute.

**(Weakness) The ablation study lacks the comparison of the feature extraction methods.**
The proposed method essentially combines the velocity, pose, and deep features for a better video anomaly detection, and only one existing model for each feature is considered throughout the paper. However, there are different approaches for these features, such as point tracking (instead of optical flow, which cannot model the occlusions) [1, 2] and video backbones (instead of CLIP, which only models the per-frame feature) [3, 4]. Since video anomaly detection fundamentally requires an understanding of object interactions (including occlusions) and video representation learning, the reviewer believes it is important to consider these backbones in the proposed method.

**(Weakness) Important details are missing in the experiment and discussions.**
- While the authors argue that replacing kNN with k-means provides a faster alternative, its runtime speed is not presented.
- Most baselines are missing the macro AUROC results in Table 1. However, these results could be filled in for those baselines that provide the open source implementation and checkpoints. For example, the official repository for the latest baselines considered in Table 1 can be readily found by searching in Google, e.g., [5, 6].
- What are the ground truth statistics for the benchmark? For example, it would be useful to present the duration, the proportion of anomalous frames in each video. Since the benchmarks are run with only 12 / 21 / 107 test scenes, it would also be useful to present discussions per scene (possibly in the appendix).

**(Weakness) Some claims need more support and description.**
- On page 3, it is claimed that "Since determining true velocity requires 3D understanding, we represent it using optical flow". However, this reviewer does not fully understand how optical flow can provide 3D understanding, since it only provides frame-by-frame 2D translation of pixels.

[1] Doersch et al., "TAPIR: Tracking Any Point with per-frame Initialization and temporal Refinement", ICCV 2023

[2] Karaev et al., "CoTracker: It is Better to Track Together", ECCV 2024

[3] Wang et al., "VideoMAE V2: Scaling Video Masked Autoencoders with Dual Masking", CVPR 2023

[4] Li et al., "VideoMamba: State Space Model for Efficient Video Understanding", ECCV 2024

[5] Hirschorn et al., "STG-NF"; https://github.com/orhir/STG-NF

[6] Cao et al., "FB-SAE"; https://campusvad.github.io/

---

> ### Author Response · Authors · 2024-10-06
>
> Thank you for your thorough review of our paper. We are glad that the reviewer found our method to be both simple and effective, and we appreciate the reviewer’s acknowledgment of the state-of-the-art performance achieved using off-the-shelf models for feature extraction. Below, we address the reviewer’s concerns point by point:
>
> 1. **Motion attribute design:** We acknowledge the limitations of optical flow in complex scenarios. Our focus on optical flow was driven by its simplicity and effectiveness in object-centric anomaly detection, which is prevalent in the tested benchmarks. While ShanghaiTech does include some multiple object interactions (e.g., violence anomalies), we agree that future work could explore more advanced motion modeling techniques and when more advanced datasets are available.
>
> 2. **Ablation study and feature extraction comparison:** In our ablation study, we focused on demonstrating the importance of each feature in our current setup (velocity, pose, and deep features). As we discussed in the Appendix, in preliminary experiments, we found that features extracted by pretrained video encoders did not work as well as pretrained image features on the type of benchmark videos used in VAD. We hypothesize that the weaker performance of video encoders may be due to their focus on capturing temporal dynamics on frame-level, whereas our approach is more object-centric. In addition, when we tested video encoders on 10-frame windows of fixed object bounding boxes (centered around time $t$), we observed no performance gain, likely due to resolution constraints and the need for high-quality contextual information. We revised the manuscript by moving and extending this discussion from the appendix to the main paper (Section 4.5) and including a table (Table 8) that summarizes our findings regarding the weak performance of video encoders.
>
>
> 3. **Missing experimental details:** (i) We added the runtime speed comparison between $k$NN and $k$-means, which can now be found in Table 5 of the revised manuscript. (ii) Regarding the missing macro AUROC results for some baselines, while we agree that it would be valuable, many VAD approaches do not provide open-source code. Nevertheless, for the baselines mentioned by the reviewer, we revised the paper and filled in the missing entries by running the available open-source implementations. These updated results are presented in Table 2 of the revised manuscript. (iii) We also added the requested ground truth statistics in Table 1 and included a per-scene discussion and analysis in the appendix (Table 9).
>
>
> 4. **Optical flow and 3D understanding:** Our intention was not to suggest that optical flow provides a full 3D understanding of the scene. Instead, our goal was to emphasize that optical flow, while fundamentally a 2D technique, offers a useful proxy for capturing motion in the absence of depth information. We revised the relevant section of the paper to clarify this point and avoid misunderstandings.

---

### Review · Reviewer_GfMb · 2024-10-02

**Summary Of Contributions:**

The paper claims a new state-of-the-art in video anomaly detection (VAD) tasks, where the goal is to identify anomalous frames within a given video clip. The method is quite simple. Specifically, it is based on the following three features - (a) the velocity feature: an 8-dimensional aggregation of its optical flow map, (b) the pose feature: the normalized coordinates of $d$ keypoints, and (c) the CLIP image embedding. A per-object anomaly score is obtained via density estimation of these features, either using Gaussian mixture (for the velocity feature) and kNN (for the others). Experimental results show that the proposed method outperforms previous methods on three public benchmarks - UCSD Ped2 (99.1% AUROC), CUHK Avenue (93.7%), and ShanghaiTech Campus (85.9%) - despite its simplicity.

**Audience:**

Yes

**Broader Impact Concerns:**

As discussed in Section 6 of the paper, the technique developed in this work - VAD systems with a human focus - naturally carries potential risks of misuse: particularly in applications related to privacy concerns, e.g., for unauthorized human surveillance. The paper does suggest several technical strategies to mitigate such risks; it may be beneficial to further include a discussion about any recommendations for regulatory oversight beyond the technical aspects, for ensuring that VADs are deployed responsibly - e.g., limiting use cases to public safety, ensuring civilian oversight over the VAD system, etc.

**Claims And Evidence:**

Yes

**Requested Changes:**

- Any discussion about scenarios when the pose features are not available, e.g., for anomaly detection of general objects.
- A further ablation study about the choice of backbone networks, e.g., flow net, object detector, and deep feature extractor.
- Comparison with more recent state-of-the-art methods.
- (minor) The overall notations could be improved; for example, the current notation states that the anomaly score $s(\phi(f))$ is solely based on the feature $\phi(f)$, i.e., per-frame. In my understanding this is not true, as the the features are actually defined using the optical flow information that sees other frames as well. The notation in Eq. 2 is also confusing to me; here $s(\cdot)$ is used inside the definition of $s(\cdot)$ itself.
- (minor) Starting sentences with I.e. or E.g. looks a bit informal to me.

**Strengths And Weaknesses:**

**Strengths**
- The paper is overall easy to follow.
- The simplicity of the proposed method is appealing to me, particularly considering its high performance compared to the previous approaches. It not only offers ease of use but also provides efficiency and interpretability, e.g., the velocity and pose features are quite straightforward to analyze.
- The experimental results successfully support the effectiveness of the proposed approach, and provides a detailed component-wise analysis, e.g., for their uses of the three different features.


**Weaknesses**
- The proposed method relies on the pose features, which is only available for humans. Ablation study says that the use of pose features have a small-but-consistent effectiveness (compared to the other two) to the final performance. This raises a question about any possible alternative when such a pose features are not available.
- The proposed method highly depends on the performance of the (pre-trained) backbone feature extractors, viz., for obtaining optical flows, bounding boxes, and CLIP features. A missing ablation study here is the individual effects of the choice of these backbones. For example, could the method further advance the state-of-the-art if we just take larger backbones? Which backbone network is the most influential for the final performance?
- On the flip side of its simplicity, I found few technical significance in the proposed method; it can be essentially regarded as a straightforward combination of basic features. Additional justification for the method design, particularly addressing why this approach is unique and effective in the context of VAD, would be be helpful.
- Although the paper provides an extensive comparison with previous methods up to 2023, it lacks of comparison with more recent methods from 2024 - for example, [1].
- The ablation study was not performed on Ped2 - wondering if there is any particular reason for this.
- The paper reports that its method runs at 12FPS overall. But there is no comparison of this value with other baselines, making it hard to evaluate in terms of its efficiency.

[1] Micorek et al., MULDE: Multiscale Log-Density Estimation via Denoising Score Matching for
Video Anomaly Detection, CVPR 2024.

---

> ### Author Response · Authors · 2024-10-06
>
> We thank the reviewer for their thoughtful comments. We are pleased that they found our paper easy to follow and appreciated the simplicity and interpretability of our method while achieving state-of-the-art performance. Regarding the specific concerns:
>
> 1. **Pose features and general objects:** While pose features provide a small but consistent improvement, our method remains highly effective even without them. For example, on Ped2, where we do not use pose features due to low resolution (see implementation details), we still achieve state-of-the-art performance (99.1% AUROC) using only velocity and deep features. For non-human objects, the deep features (CLIP embeddings) implicitly capture relevant attributes besides velocity. To clarify, we included in the revised manuscript an extended discussion regarding pose features when pose features are unavailable, particularly for non-human-centric anomaly detection tasks.
>
> 2. **Backbone analysis:** We performed additional ablation studies to evaluate the impact of different backbone networks on the overall performance of our method. Specifically, we tested alternative backbones for optical flow (FlowNet2 vs. RAFT), as shown in Table 6, object detection (Mask R-CNN vs. YOLO-v8) in Table 7, and feature extraction (CLIP vs. other backbones, including video encoders) in Table 8. The results indicate that the effectiveness of our approach is primarily driven by the feature design rather than any specific choice of backbone. These findings are presented in the revised manuscript.
>
> 3. **Technical significance:** Our core contribution lies in demonstrating that explicit, interpretable attributes can outperform complex deep learning architectures in video anomaly detection. This finding challenges the prevailing trend toward increasingly sophisticated neural networks and reveals that carefully chosen semantic features (velocity and pose) capture the essence of what makes events anomalous. Furthermore, our method's simplicity is itself a significant contribution as it establishes a strong baseline against which future, more complex approaches must justify their additional complexity.
>
> 4. **Recent comparisons:** We updated the manuscript to include two CVPR 2024 papers, demonstrating that our method still outperforms more updated baselines. Thanks for pointing this out.
>
> 5. **Ablation study on Ped2:** The ablation study was initially not conducted on Ped2 due to the simple nature of the dataset and the low resolution, which prevents reliable pose feature extraction. However, as per the reviewer’s request, we have now included an ablation study for Ped2 in the revised manuscript (Table 3), using velocity and deep features only.
>
> 6. **FPS comparison with baselines:** We agree that comparing FPS with other methods would help in evaluating the efficiency of our approach. However, since most VAD methods do not report FPS and do not provide open-source code, it is challenging to make direct comparisons. A fair comparison would require all methods to be tested in the same environment. Nonetheless, as per the reviewer’s request, we have evaluated two methods on our hardware: BA-AED (Georgescu et al., 2021) achieves 24 FPS, while HF$^{2}$ (Liu et al., 2021) runs at 12 FPS. Our method, at 12 FPS, exhibits slightly slower runtime performance than BA-AED but is comparable to HF$^{2}$. We have expanded our discussion on running times in the revised manuscript and included these comparison results.
>
> 7. **Minor comments and notations:** While the anomaly score $s(\phi(f))$ is indeed computed per frame, it is true that optical flow uses a two-frame window to compute flow maps. However, the final anomaly score is still per-frame. Moreover, we revised the notations for clarity, especially in Eq. 2, and reworded the sentences starting with "i.e." and "e.g." to maintain a more formal tone throughout the paper.
>
> 8. **Broader impact discussion:** We expanded the discussion on ethical considerations in the revised manuscript. Specifically, we now address the importance of regulatory and oversight mechanisms, advocating for civilian oversight to prevent misuse, especially in sensitive contexts like law enforcement or public surveillance. Additionally, we propose restrictions on VAD deployment to ensure its use is confined to cases where public safety benefits outweigh the risks, with legal authorization required for more invasive use-cases.

---

### Author Response · Authors · 2024-10-06

We sincerely thank all reviewers for their valuable feedback and constructive comments. We are pleased that all reviewers recognized the key strengths of our work. Reviewer (zhpF) commended the clarity of our paper and the thoroughness of our experimental work, while Reviewer (z24s) highlighted the simplicity and effectiveness of our method, and acknowledged the state-of-the-art performance achieved using off-the-shelf models for feature extraction. Reviewer (GfMb) also found the paper easy to follow and appreciated the simplicity, interpretability, and strong performance of our method.

In response to the reviewers’ suggestions, we revised the manuscript, with text changes highlighted in blue to indicate modifications.

Please find detailed responses to your specific concerns below.

---

### Decision · Action_Editor_D8oZ · 2024-12-10

**Recommendation:** Accept as is

**Comment:**

The paper proposes a simple-yet-effective method for video anomaly detection (VAD). It claims that combining simple features, such as velocity, pose, and CLIP-based deep representations, is sufficient to achieve near state-of-the-art performance on VAD. The claim is validated through experiments conducted primarily on three public benchmarks.

All reviewers have acknowledged the simplicity and effectiveness of the proposed method, affirming that the paper's main claim is well-supported. Additionally, the authors successfully addressed reviewers’ comments in their revision, leading all reviewers to agree on the paper's acceptance. Therefore, AE recommends acceptance.

**Audience:**

Yes

**Claims And Evidence:**

Yes

---

> ### Author Response · Authors · 2025-01-15
>
> We sincerely thank the area chair and all reviewers for their time and effort invested in reviewing our paper.
> Below is a summary of the main updates we made to the camera ready:
> * We performed an additional spell check and revised the appendix to avoid repetitive content.
> * We included an acknowledgment section.
> * We deanonymized the paper and provided a link to our official code.
>
> Once again, we sincerely thank the area chair and all the reviewers!